# Retinal chromophore charge delocalization and confinement explain the extreme photophysics of Neorhodopsin

Riccardo Palombo[1,2], Leonardo Barneschi[1], Laura Pedraza-González[1], Daniele Padula [1], Igor Schapiro [3] & Massimo Olivucci [1,2] ✉

The understanding of how the rhodopsin sequence can be modified to exactly modulate the spectroscopic properties of its retinal chromophore, is a pre-requisite for the rational design of more effective optogenetic tools. One key problem is that of establishing the rules to be satisfied for achieving highly fluorescent rhodopsins with a near infrared absorption. In the present paper we use multi-configurational quantum chemistry to construct a computer model of a recently discovered natural rhodopsin, Neorhodopsin, displaying exactly such properties. We show that the model, that successfully replicates the relevant experimental observables, unveils a geometrical and electronic structure of the chromophore featuring a highly diffuse charge distribution along its conjugated chain. The same model reveals that a charge confinement process occurring along the chromophore excited state isomerization coordinate, is the primary cause of the observed fluorescence enhancement.

Modern neuroscience requires membrane-localized signaling tools[1,2] that could emit intense fluorescence upon irradiation with red light. However, until recently, the available tools, based on engineered microbial rhodopsins, could only generate weak fluorescence signals that impair their performance. At the molecular level, the optical properties of microbial rhodopsins owe to the presence of a covalently bounded all-*trans* retinal protonated Schiff base (rPSB) chromophore and its interaction with the surrounding protein environment. Therefore, a deep molecular comprehension of the factors dictating such properties is highly desirable. In this regard, few studies[3–5] have formulated rules for tailoring the absorption and emission properties of the retinal chromophore based on the effects of homogeneous electrostatic fields acting on isolated chromophores or via chromophore chemical modifications. However, it is expected that a simple electrostatic picture could not be sufficient to explain the origin of these properties in the complex environment offered by the protein cavity since other factors like non-homogeneous electrostatic fields or chromophore-cavity steric effects could play an important role.

In 2020 the discovery of Neorhodopsin (NeoR) offered an unprecedent case study that could potentially expand our comprehension of red-shifted and highly fluorescent rhodopsins. NeoR is a rhodopsin guanylyn-cyclase (RGC) expressed in the *Rhizoclosmatium globosum* from Chytridiomycota, the only phylum of fungi producing motile and flagellated spores (zoospores)[6,7]. It heterodimerizes with other two RGCs, called RGC1 and RGC2, that have sensitivity in the blue-green spectrum with 550 and 480 nm absorption maxima ($\lambda^a_{max}$), respectively. In contrast, NeoR displays the strongest bathocromic shift among all known microbial rhodopsins, yielding an extremely red-shited ($\lambda^a_{max} = 690$ nm) absorption band. Such a band is mirrored by an intense emission band with a maximum ($\lambda^f_{max}$) at 707 nm yielding Stokes shift of only 17 nm (350 cm$^{-1}$). The emission brightness is quantified by a fluorescence quantum yield (FQY) of 20% and by an extinction coefficient ($\epsilon$) of 129,000 M$^{-1}$ cm$^{-1}$. In addition, the excited state lifetime (ESL) of 1.1 ns points to a slow excited state deactivation. The FQY of NeoR, only ca. four times weaker than that of the green fluorescent protein[8] (GFP), represents an anomaly in the rhodopsin superfamily and suggests an evolution-driven origin. More specifically,

[1]Dipartimento di Biotecnologie, Chimica e Farmacia, Università di Siena, via A. Moro 2, I-53100 Siena, Italy. [2]Department of Chemistry, Bowling Green State University, Bowling Green, OH 43403, USA. [3]Fritz Haber Center for Molecular Dynamics, Institute of Chemistry, The Hebrew University of Jerusalem, 9190401 Jerusalem, Israel. ✉e-mail: olivucci@unisi.it

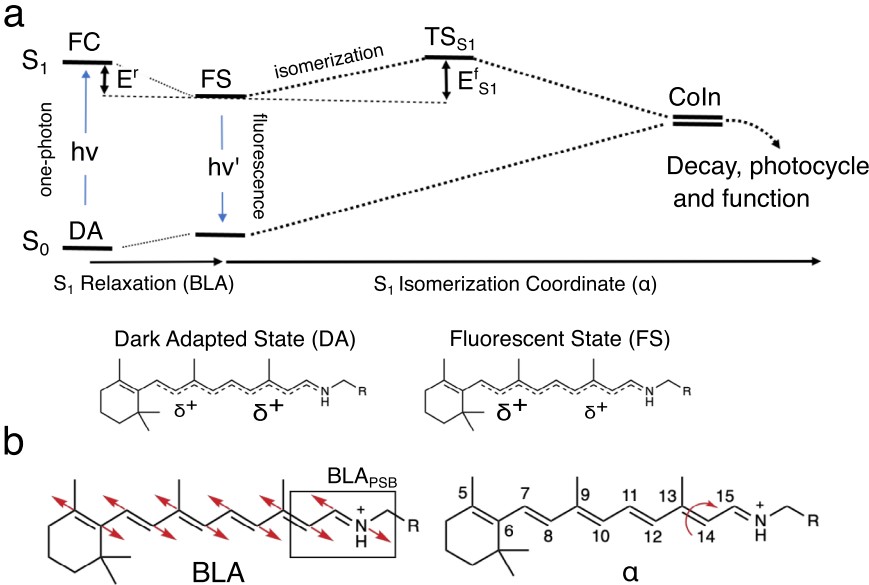

**Fig. 1 | Geometrical and electronic structure changes in NeoR. a** Schematic representation of the hypothetic $S_0$ and $S_1$ energy changes occurring along the $S_1$ relaxation that involves the bond length alternation (BLA, quantified by the difference between the average of the double-bond lengths and the average of the single-bond lengths of the conjugated chain) and isomerization ($\alpha$) coordinates. The rPSB resonance hybrids show a delocalized positive charge at the $S_0$ and $S_1$ energy minima corresponding to the Dark Adapted State (DA) and Fluorescent State (FS), respectively. The symbol "$\delta$+" gives a qualitative measure of the amount of positive charge located along the rPSB-conjugated chain. **b** Representation of the bond length alternation (BLA) mode and the torsion mode ($\alpha$) along the C13=C14 double bond. $BLA_{PSB}$ is the -C14-C15 and C15-N bond lengths difference.

since the emission competes with the photoisomerization of its rPSB chromophore, a presently unknown adaptation process must have decreased the efficiency of the protein function. This hypothesis is in line with the fact that wild-type (WT) rhodopsins commonly exhibit FQYs spanning the 0.0001%–0.01%[9–12] range while engineering efforts yielded variants with only modest increases up to a 1.2% value[13–16].

Deciphering how natural evolution in NeoR has tuned these extreme spectroscopic properties of the rPSB chromophore could expand our ability to design optogenetic tools with augmented functionality. Therefore, the modeling of NeoR represents a new promising learning opportunity that can be also used to assess the transferability of the rules mentioned above. In particular, NeoR offers the opportunity to disclose the molecular-level mechanism controlling the branching between fluorescence emission and photoisomerization. Such branching, which is schematically illustrated in Fig. 1a for all-*trans* rPSB, has been shown to dominate the fluorescence modulation in a set of GFP-like protein variants[8,17]. More specifically, in these systems, the FQY appears to be directly proportional to the energy barrier ($E^f_{S1}$) controlling both access to a conical intersection (CoIn) located along the first singlet excited state ($S_1$) isomerization coordinate and the decay to the ground state ($S_0$). Here we assume that the same mechanism operates in NeoR is then used as a "laboratory" model for proposing a mechanism capable to connect sequence variation and rPSB emission. To do so, we also assume, in line with the evidence coming from a set of Arch3 variants displaying enhanced fluorescence[18,19], that the NeoR emission is a one-photon process and that, therefore, originates directly from its dark adapted state (DA).

In order to pursue the objectives above, we construct a quantum-mechanics/molecular mechanics (QM/MM) model of NeoR based on multiconfigurational quantum chemistry. Since a crystallographic structure is not available, we employ, for the model construction, a previously reported comparative model[7]. While this may limit, in principle, the fidelity of the environment description with respect to that found in nature, our target here is to achieve an atomistic model capable to replicate all relevant spectroscopic and photochemical observables and use it to explain the high FQY of NeoR in terms of geometrical, electrostatic and steric effects.

Accordingly, here we firstly use the QM/MM model to investigate the electronic structures of the NeoR DA and fluorescent state (FS) and, secondly, we use it to investigate the NeoR photoisomerization with the target of documenting the magnitude and origin of $E^f_{S1}$. We show that the confinement of the delocalized positive charge on the Schiff base moiety of the rPSB backbone can explain the existence of large $E^f_{S1}$ values and, in turn, the high FQY of NeoR.

## Results and discussion
### Model construction and validation
An initial QM/MM model of NeoR was constructed using the Automatic Rhodopsin Modeling (*a*-ARM) technology[20–22] starting from the comparative model mentioned above. *a*-ARM models have been shown to yield congruous (i.e., built by employing exactly the same protocol) animal and microbial rhodopsin models that correctly reproduce trends in $\lambda^a_{max}$ values[9,20,21,23–27]. The model showed that the NeoR all-*trans* rPSB is embedded in a cavity featuring a peculiar amino acid composition with two glutamic (E136 and E262) and one aspartic (D140) acid residues located in the vicinity of its Schiff base moiety. However, due to the lack of experimental information on the residue protonation state, the chromophore counterion assignment remains ambiguous[28]. For this reason, a set of customized *a*-ARM models featuring different protonation states for the E136, E262, D140 plus E141, a residue located halfway along the rPSB conjugated chain (see Fig. 2a), were built and ranked by computing the absorption ($\lambda^a_{max}$) and emission ($\lambda^f_{max}$) maxima as well as the relaxation energy ($E^r$) defined by the basic mechanism of Fig. 1a. The $\lambda^a_{max}$ and $\lambda^f_{max}$ values were computed in terms of vertical excitation energies ($\Delta E_{S0–S1}$) between $S_0$ and $S_1$ at the DA and FS equilibrium geometries, respectively. $E^r$ was instead computed as the energy difference between the Franck–Condon (FC) point and FS state and, therefore, quantifies the energy decrease associated with $S_1$ relaxation. The results collected in Fig. 2b that display the $\lambda^a_{max}$, $\lambda^f_{max}$, and $E^r$ values for models where the "counterion tetrad" defined above have total charges of 0, −1, −2, −3. For completeness, we have also reported the scenario with a total charge −3 even if the two transitions displayed by these models are not allowed, being the oscillator strengths ($f_{S0–S1}$) close to zero.

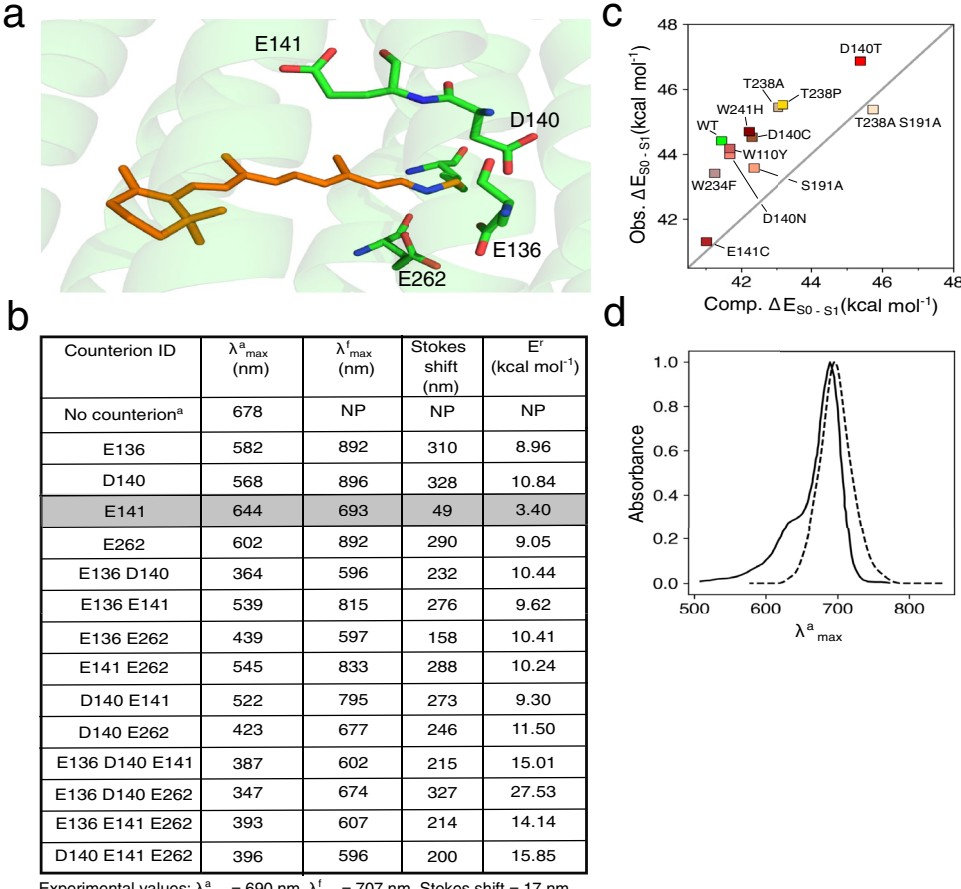

| Counterion ID | $\lambda^a_{max}$ (nm) | $\lambda^f_{max}$ (nm) | Stokes shift (nm) | $E^r$ (kcal mol$^{-1}$) |
|---|---|---|---|---|
| No counterion[a] | 678 | NP | NP | NP |
| E136 | 582 | 892 | 310 | 8.96 |
| D140 | 568 | 896 | 328 | 10.84 |
| E141 | 644 | 693 | 49 | 3.40 |
| E262 | 602 | 892 | 290 | 9.05 |
| E136 D140 | 364 | 596 | 232 | 10.44 |
| E136 E141 | 539 | 815 | 276 | 9.62 |
| E136 E262 | 439 | 597 | 158 | 10.41 |
| E141 E262 | 545 | 833 | 288 | 10.24 |
| D140 E141 | 522 | 795 | 273 | 9.30 |
| D140 E262 | 423 | 677 | 246 | 11.50 |
| E136 D140 E141 | 387 | 602 | 215 | 15.01 |
| E136 D140 E262 | 347 | 674 | 327 | 27.53 |
| E136 E141 E262 | 393 | 607 | 214 | 14.14 |
| D140 E141 E262 | 396 | 596 | 200 | 15.85 |

Experimental values: $\lambda^a_{max}$ = 690 nm, $\lambda^f_{max}$ = 707 nm, Stokes shift = 17 nm
[a] This model does not exhibit a stable FS

**Fig. 2 | Choice of the NeoR chromophore counterion and model assessment.**
**a** Overview of the structure of the all-trans rPSB chromophore (orange) and its four potential residue counterions (in green). The lysine residue (in green) bounded to the rPSB chromophore is also displayed. **b** Computed (CASPT2 level) maximum absorption wavelength ($\lambda^a_{max}$), maximum emission wavelength ($\lambda^f_{max}$) and relaxation energy ($E^r$) of NeoR with varying counterion choices. **c** Correlation between experimental (Obs. $\Delta E_{S0-S1}$) and computed (Comp. $\Delta E_{S0-S1}$) values of vertical excitation energies defining $\lambda^a_{max}$ in the wild type (indicated as WT) and a set of NeoR mutants. **d** Superimposition of experimental and computed (dotted line) absorption band of wild type NeoR. The experimental band has been digitalized from the corresponding ref. 7.

Figure 2b shows that the model with a deprotonated E141 and neutral E136, D140, and E262 (from now on $a$-ARM$_{E141}$) is the most accurate. In fact, $a$-ARM$_{E141}$ yields $\lambda^a_{max}$ and $\lambda^f_{max}$ values only 46 and 14 nm blue-shifted with respect to the experimental value as well as the smallest difference (49 nm) between those values consistent with the tiny Stokes shift experimentally observed for NeoR. All other assessed protonation states yielded a poor comparison with the observed quantities. For instance, although the model with all four residues protonated (i.e., with no counterion) produces a $\lambda^a_{max}$ close to the experimental one, it lacks a stable FS structure since no energy barrier could be located preventing access to the CoIn along $\alpha$. Also, consistently with the high intensity of the observed absorption and emission bands[7], the $a$-ARM$_{E141}$ computed oscillator strengths are found to be very high (see Supplementary Tables 1 and 2): 1.71 and 1.90, respectively. Such values were confirmed via multistate XMCQDPT2 calculations that yielded values close to 1.66 and 1.80.

In order to further assess the quality of $a$-ARM$_{E141}$, we constructed the models of a set of NeoR variants whose $\lambda^a_{max}$ values have been experimentally measured (see Supplementary Table 3). As shown in Fig. 2c the models reproduce the observed trend indicating that a-ARM$_{GLU141}$ describes, qualitatively, the effect of cavity residue replacements. Notice that the trend is reproduced with a systematic blue shift, which is typical of $a$-ARM models[20,21,23,25–27,29]. We also used

$a$-ARM$_{E141}$ to simulate the WT NeoR absorption band at room temperature by computing the $\Delta E_{S0-S1}$ and S$_0 \rightarrow$S$_1$ transition probability values for 200 snapshots representing the Boltzmann distribution (see Supplementary Section 3). Comparison between the simulated and observed data in Fig. 2d shows that the center of the computed band (703 nm) is only 13 nm red-shifted with respect to the experimental $\lambda^a_{max}$ value and the computed band half-width is close to that seen experimentally. $a$-ARM$_{GLU141}$ only appears to miss a shoulder at 640 nm that is, likely, of vibronic origin[7] and therefore not captured by a simulation based on the Condon approximation.

### Electronic character of the DA and FS vertical transitions

The agreement between experimental and computed data allows to use $a$-ARM$_{E141}$ to investigate the large bathochromic shift, negligible Stokes shift, and intense fluorescence of NeoR. The aim of Fig. 3b, c is to document the variation in electronic character upon light absorption and emission by looking at the Mayer bond order analysis (see Supplementary Table 6) and vertical electron density changes at DA and FS ($\delta\rho_{abs}$ and $\delta\rho_{emi}$, respectively). The results support the hypothesis that *both* the S$_1$ and S$_0$ electronic characters are combinations of putative covalent (COV) and charge transfer (CT) diabatic states loosely associated with the limiting resonance structures of Fig. 3a. In particular, the results reveal that, the DA structure displays an unusually large CT weight in S$_0$ yielding a positive charge spread

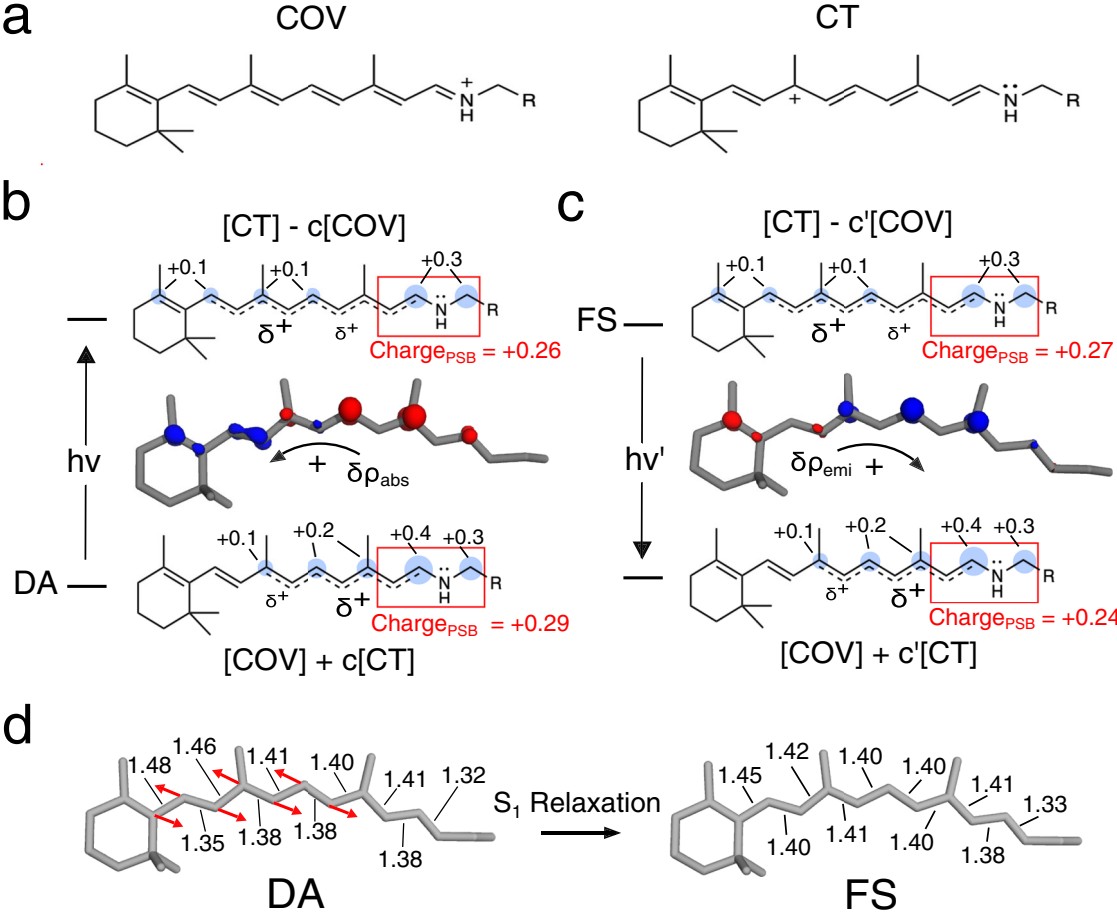

**Fig. 3 | Electronic and geometrical character of $S_1$ relaxation in NeoR chromophore. a** Representation of the two limiting resonance formulas adopted to describe the electronic character of the rPSB chromophore. **b** Electron density variation ($\delta\rho_{abs}$) characterizing the vertical $S_0 \to S_1$ transition from the Dark Adapted State (DA). Blue and red clouds correspond to electron density decrease and increase respectively. Isovalue set to 0.002 a.u. The associated resonance formulas correspond to resonance hybrids also anticipated in Fig. 1a. Blue bubbles represent the QM positive charge (in e unites). Only absolute values > 0.05 e are reported. As indicated by the red box, the total charge residing in the -C14-C15-N-Cε- rPSB fragment is also given. **c** Same data for the $S_1 \to S_0$ emission from the Fluorescent State (FS). **d** Geometrical comparison between DA and FS rPSB structures. The arrows indicate the dominant geometrical change corresponding, clearly, to a variation in the bond length alternation (BLA, see definition in the caption of Fig. 1) in a region of the conjugated chain distant from the Schiff base moiety. The relevant bond lengths are given in Å.

along the carbon atoms of the chromophore (see the bubble representation from Mulliken charges) with a limited +0.29 e charge residing in the C14-C15-N-Cε moiety (from now on, the charge residing on such moiety will be called Charge$_{PSB}$). The vertical transition to $S_1$ only slightly modifies such charge distribution. For instance, when taking the mid C13=C14 as a reference one can see only a small 0.03 e translocation towards the β-ionone ring. The same behavior is seen at the FS when looking at the vertical emissive transition for which one finds Charge$_{PSB}$ values of +0.27 e and +0.24 e for $S_1$ and $S_0$, respectively. In conclusion, as illustrated in Fig. 3b, c, the DA and FS transitions can be both qualitatively interpreted as transitions between adiabatic states (i.e., $S_0$ and $S_1$) corresponding to in-phase and out-of-phase mixing of two diabatics (or resonance formulas) close in energy. Such an interpretation appears to be related to the one proposed for explaining the observed absorption and emission trends of GFP-like proteins[17].

The description above is not in line with the consensus electronic structure of the rPSB chromophore[30,31] of rhodopsins. In fact, the DA $S_0 \to S_1$ transition, is usually described as a transition starting from a COV-dominated state featuring a positive charge localized on the -C15=N- moiety and not a delocalized charge spread on the -C9=C10-C11=C12-C13=C14-C15=N- chain as seen in Fig. 3b. Starting from such a

state, an at least three times larger charge translocation has been computed upon $S_0 \to S_1$ excitation[32,33].

Notice that the chromophore charge delocalization seen in the selected $a$-ARM$_{E141}$ model is modulated by the position of the counterion. In fact, the charge distribution of the model featuring E262 as the only charged residue of the tetrad (see Fig. 2b) features, in the DA state, a blue-shifted $\lambda^a_{max}$ and a reduced charge delocalization. These values are accompanied by a much larger +0.68 e to +0.27 e change in Charge$_{PSB}$ value upon vertical excitation (see Supplementary Fig. 2) and are, therefore, more in line with the consensus rPSB charge distribution mentioned above.

**FC→FS geometrical and electronic relaxation**

Consistently with the computed negligible (0.01 e) difference in Charge$_{PSB}$ value between the FC point and the FS state, the $S_1$ electronic relaxation of $a$-ARM$_{E141}$ can be interpreted as a relatively minor change in the weights of the COV and CT diabatic states. The geometrical variation accompanying such a process is documented in Fig. 3d and corresponds to a minor progression along the BLA coordinate (this is defined as the difference between the average single-bond length and the average double-bond length of a conjugated chain, see Fig. 1b) of the chromophore, leading to an $E' \approx 3.5$ kcal mol$^{-1}$

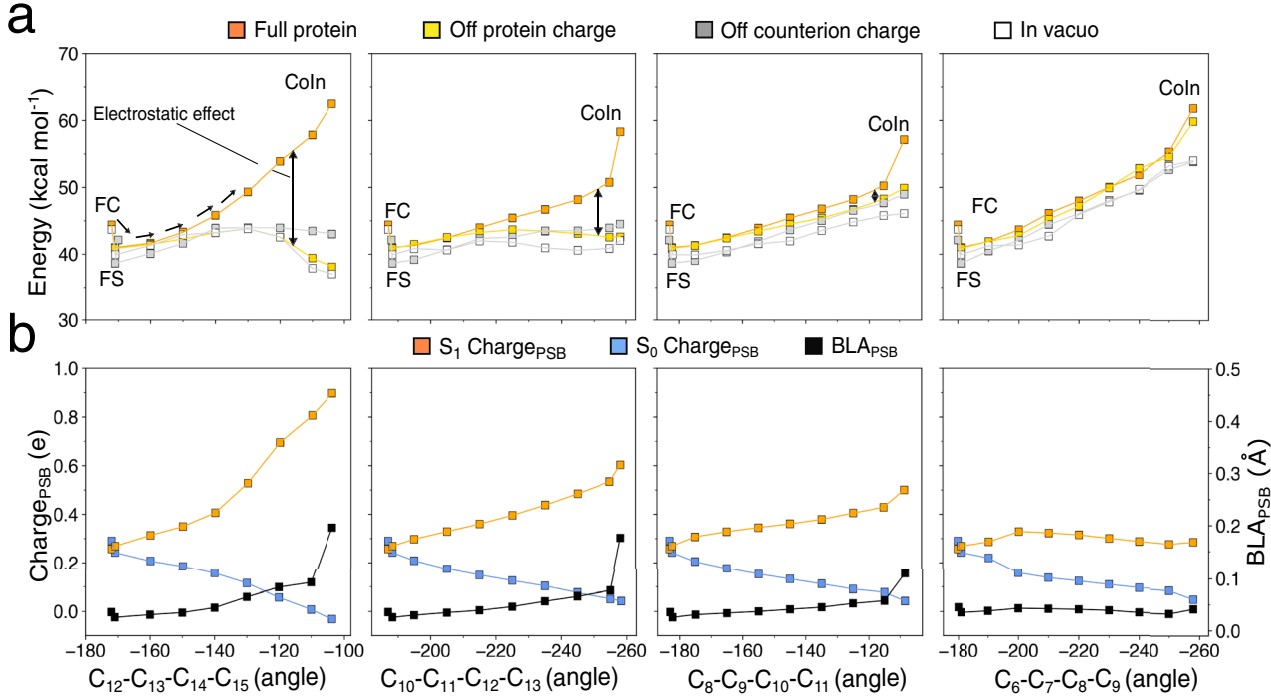

**Fig. 4 | $S_1$ energy, charge and BLA profiles along the photoisomerization of, from left to right, C13=C14, C11=C12, C9=C10 and C7=C8 double bonds.** **a** CASPT2 $S_1$ energy profiles computed in presence of the full protein environment (orange squares), after setting to zero the MM charges of the entire protein (gold squares), after setting to zero only the MM charges of the rPSB counterion (i.e., E141, gray squares) and after removing the whole protein in a full QM calculation (i.e., in vacuo, empty squares). **b** Evolution of $S_0$ and $S_1$ Charge$_{PSB}$ (the charge residing in the C14-C15-N-Cε rPSB moiety) and evolution of BLA$_{PSB}$ (C14-C15 and C15-N bond lengths difference, see Fig. 1b).

value (see Fig. 2b). Such assignment is based on the hypothesis of a negligible contribution from the surrounding protein environment, in line with the results reported for GFPs[17]. Interestingly, the BLA-driven nature of the $S_1$ relaxation is a known feature among GFPs where the only geometrical difference between the DA and FS states is mainly attributed to a different BLA displacement[17]. In NeoR the BLA change is mostly described by a contraction and elongation of C6-C7/C8-C9/C10-C11 and the C7=C8/C9=C10/C11=C12 bonds respectively, while the BLA of the C14-C15-N Schiff base moiety (from now on simply BLA$_{PSB}$, see Fig. 1b) is minimally involved and only changes of 0.01 Å.

**NeoR photoisomerization**

As anticipated above, here we assume that the branching (see Fig. 1a) between the canonical rPSB double bond photoisomerization of microbial rhodopsin and emission determines the fluorescence brightness. In other words, since the $S_1$ isomerization is intrinsically a non-radiative decay process, the $E^f_{S_1}$ barrier controlling its rate must modulate the ESL and, in turn, the FQY value. Accordingly, we have used a-ARM$_{E141}$ to compute the approximate $S_1$ minimum energy path (MEP) describing the clockwise (CW) torsional deformation along $\alpha$, namely the C12-C13-C14-C15 dihedral angle describing the C13=C14 double bond torsion connecting the FS to the ca. 90° twisted conical intersection (CoIn) giving access to $S_0$. The energy maximum located along the MEP energy profile must therefore reflect the barrier height (i.e., $E^f_{S_1}$).

Given the current uncertainty about the regiochemistry of the photoisomerization reaction in NeoR[34,35], we also computed three additional MEPs (see Supplementary Section 5) corresponding to the counterclockwise (CCW) isomerization of the C11=C12 and C7=C8 double bond and to the CW isomerization of the C9=C10 double bond. The choice of the CW/CCW pattern for adjacent double bonds conforms to the well-known aborted bicycle-pedal motion[30,36], the archetypal space-saving reaction coordinate for the rPSB chromophore

isomerization. The selected direction of the twisting is consistent with the stereochemistry found in the four computed CoIn's where the constrained rotation of the reactive bond is assisted by the opposite rotation of the adjacent double bonds (see Supplementary Fig. 3).

Figure 4 displays the CASPT2 $S_1$ energy profiles (top panels) together with the evolution of the $S_0$ and $S_1$ Charge$_{PSB}$ and the BLA$_{PSB}$ coordinate (bottom panels). As reported in the previous works[30,32,37], and also above, such charges are used as indicators of the weights of the COV and CT diabatic states in the adiabatic $S_0$ and $S_1$ energy profiles. As detailed in section S5, the energy trends that emerged from the CASPT2 level were also confirmed at the XMCQDPT2 level. All four $S_1$ isomerization paths point to the presence of a barrier (see orange energy profiles). However, the $S_1$ energy profile is qualitatively different from the one hypothesized in Fig. 1a. In fact, in all cases the energy increases monotonically from the FS state with the CoIn corresponding to the highest point along the MEP. The plot is consistent with a sloped, rather than peaked, topography of the ca. 90° twisted CoIn structures[38] (see also the branching plane map in Supplementary Fig. 13 for the CoIn located along the C13=C14 MEP). As shown in Supplementary Fig. 5, we estimated the $E^f_{S_1}$ magnitude as the difference in energy between the CoIn and the FS states. $E^f_{S_1}$ is found of 21 (25), 17 (22), 16 (16), 20 (16) kcal mol$^{-1}$ at the CASPT2 (XMCQDPT2) level for respectively the C13=C14, C11=C12, C9=C10 and C7=C8 isomerizations. Since the lowest energy barrier is still relatively large, our model supports a barrier-controlled mechanism for the FS emission. Thus, NeoR would be an analog of the GFP-like fluorescent reporters[8,30] as, in contrast with microbial rhodopsins such as bacteriorhodopsin[30], its relatively high barrier would induce slow internal conversion kinetics. This conclusion is consistent with the nanosecond ESL of NeoR while, most known fluorescent rhodopsins feature an ESL that does not exceed[13,30,39-41] the picosecond range.

Notice that the model predicts lower energy barriers with respect to the canonical microbial C13=C14 isomerization which appears

highly disfavored. Hence, alternative photoisomerization pathways might originate as recently observed by Sugiura et al.[35] in the NeoR from *Obelidium mucronatum* (OmNeoR) that shares 78% of sequence identity with the NeoR studied in this work. This result appears consistent to what was found in the past by Cembran et al.[3] aimed at investigating the relationship between the position of an acetate counterion and the photoisomerization of a nearly isolated protonated polyene chain. They found that placing the counterion above the polyene favors the isomerization of the double bonds closest to the counterion; this result can be loosely associated with the favored C11=C12, C9=C10 and C7=C8 isomerization in our a-ARM$_{E141}$ model. However, we need to stress that the obtained $E^f_{S1}$ values possibly represent upper limits as a sloped CoIn features, in its close vicinity, a slightly lower energy region with the same 90° twisted conformation (see Supplementary Fig. 13).

In order to check the existence of alternative and lower energy photoisomerization channels, we computed the C8-C9, C10-C11, and C12-C13 single bond MEPs and found that in all cases $E^f_{S1}$ is >20 kcal/mol$^{-1}$ (see Supplementary Figs. 9 and 10). This supports the high stability of the FS displayed by our a-ARM$_{E141}$ model.

The evolution of the Charge$_{PSB}$ along the C13=C14 isomerization coordinate (see the orange curve in the corresponding panel of Fig. 4) reveals that the S$_1$ weight of the COV diabatic state increases monotonically along the S$_1$ MEP until it dominates the region approaching the CoIn. This corresponds to confinement (or localization) of the charge in the small Schiff base moiety that, along the terminal part of the MEP (i.e., near the 90° twisted conformation), hosts a π-system orthogonal to the one residing along the rest of the rPSB conjugated chain. This is not a general behavior that depends on the isomerizing double bond. In fact, the C7=C8 MEP in Fig. 4 appears to feature, along the entire S$_1$ profile, a steady mixed COV/CT character. These results point to a change in the origin of the critical $E^f_{S1}$ barrier along different isomerization coordinates. More specifically, it is expected that the electrostatic effect imposed by the NeoR cavity may have different effects along different MEPs with a maximal effect on the canonical C13=C14 energy profile and a minimal effect on the C7=C8 energy profile. It is thus necessary to also evaluate steric effects.

In our ARM$_{E141}$ model the electrostatic effect is due to the protein point charges including those describing the negatively charged E141 counterion. In order to assess the impact of such an effect on the isomerization energy profiles, these have been re-evaluated after setting to zero all protein point charges while keeping the geometrical progression unchanged (see gold energy profiles). Consistently, with a dominant role played by the protein electrostatics, the slope in the S$_1$ profile associated with the C13=C14 coordinate is strongly decreased and even inverts from positive to negative in its last part. This effect is gradually reduced along the C11=C12 and C9=C10 coordinates (compare the vertical double arrows) and disappears along the C7=C8 coordinate of Fig. 4d. The energy profiles were also re-computed after the removal of the E141 counterion charge exclusively (see gray energy profiles). When compared to the energy progression seen in absence of the protein electrostatics, the effect is reduced but maintained, indicating that the leading electrostatic contribution is due to the negative charge in the E141 position. This behavior is consistent with the lack of a stable FS (i.e., due to the absence of an S$_1$ energy barrier controlling access to the CoIn) displayed by the model with no counterions (see Fig. 2b). In fact, switching off the charge of the E141 counterion roughly replicates the electrostatic embedding imposed by that model.

The models featuring a counterion configuration different from that of a-ARM$_{E141}$, display flat, and substantially barrierless, S$_1$ isomerization energy profiles (see Supplementary Fig. 11). As stated above, this is not consistent with the ESL of NeoR estimated to be 1.1 ns. Therefore, our data indicate that an E141 counterion appears not only critical for tuning the extreme spectroscopy of NeoR (see Fig. 2b) but also for the generation of a barrier.

To disentangle the electrostatic and steric contributions to the computed $E^f_{S1}$ value, the same energy profiles have been re-evaluated in the absence of a whole protein environment (see the energy profiles marked with empty squares). The results demonstrate that while in C11=C12 and C9=C10 MEPs the S$_1$ profile becomes completely flat, in C7=C8 MEP the S$_1$ energy barrier is only reduced but persists, indicating a destabilization that originates from the rPSB geometrical progression. Notice that such progression is due, in all cases, to indirect electrostatic and steric effects determining the DA, FS, and CoIn geometries (i.e., determining the isomerization coordinate) and include the effect of the polarization of the rPSB π-electron density due to the counterion.

Our conclusion is that a small barrier increasing along the C13=C14 to C7=C8 series, is an intrinsic feature of the isomerization coordinate computed using a-ARM$_{E141}$. While such an increase is clearly enhanced when switching on the *direct* steric interactions (i.e., due to the Lennard-Jones potentials between QM and MM atoms) are considered, the $E^f_{S1}$ value along the C13=C14 and C11=C12 paths remain flat, and inconsistent with bright emission. To enhance these barriers and restore consistency, a direct electrostatic contribution (i.e., due to the interaction between QM electron density and MM point charges) is critical. In the next section, we look at the mechanism driving such a critical electrostatic effect.

## Fluorescence enhancement mechanism

In microbial rhodopsins the canonical S$_1$ isomerization produces the 13-*cis* rPSB chromophore. In general, this is an ultrafast (sub-picosecond) reaction only allowing a negligible fluorescence emission from the DA state. We now use the results above to formulate a mechanistic theory for the fluorescence enhancement explaining how the NeoR electrostatics generates the high C13=C14 isomerization barrier of Fig. 4. Such theory takes the progressive confinement of the initially delocalized rPSB charge described above as the key event blocking the C13=C14 isomerization.

We start by employing the Charge$_{PSB}$ and BLA$_{PSB}$ quantities defined above to follow the chromophore geometrical and electronic changes along the S$_1$ isomerization coordinate. The first index displays a monotonic charge increase from 0.27 to 0.90 e (at FS and CoIn respectively), consistently with a monotonic increase of the positive charge on the Schiff based chromophore moiety. The second index points to a 0.03 to 0.20 Å change consistently with the reconstitution of a C=N double bond along the path and full localization of the charge on such a bond. Such progressive charge confinement is directly proportional to the increase in the electrostatic effect along the MEP of C13=C14 of Fig. 4 (i.e., the one indicated by the double-headed vertical arrow) and, therefore, to the energy increase leading to the large computed $E^f_{S1}$ value. We now propose that the molecular mechanism driving the energy increase is the progressive increase in distance between the negative E141 counterion charge and the centroid of the confining charge. As illustrated in Fig. 5a the progressive positive charge confinement shifts the centroid of the positive charge away from the E141 residue, unavoidably leading to destabilization. This mechanism is supported by the computed decrease in electrostatic effect (i.e., again, the destabilization indicated by the vertical arrow) along the C13=C14, C11=C12 and C7=C8 MEPs of Fig. 4. As an example, in Fig. 5a we also show that the C9=C10 isomerization could not lead to the same electrostatic effect as, in this case, the charge does not get confined far from E141 but remains delocalized along the extended C10-C11-C12-C13-C14-C15-N moiety. This causes only a limited change in the counterion-chromophore interaction consistently with the computed decrease in electrostatic stabilization. The charge confinement on the Schiff base moiety is thus critical.

The mechanism described above can be reinterpreted in terms of changes in the energy of the COV diabatic state (see H$_{COV}$ in Fig. 5b) featuring a positive charge permanently located on the Schiff

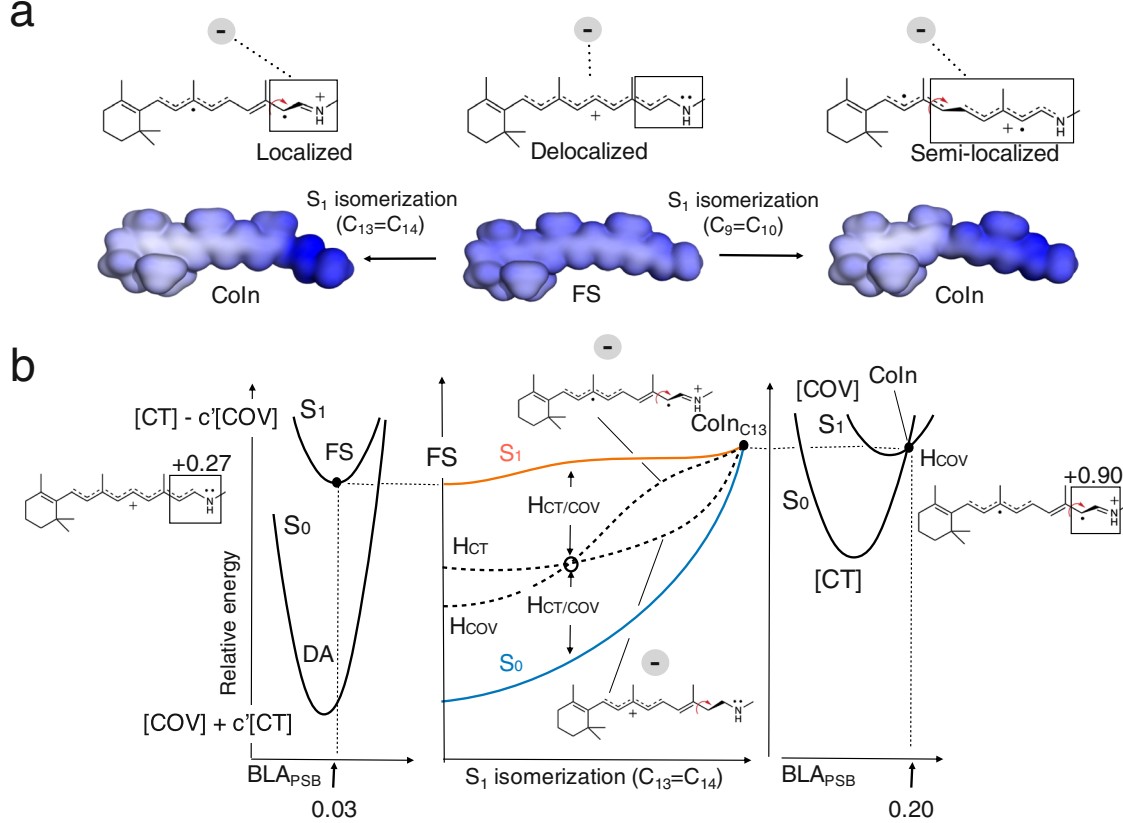

**Fig. 5 | Origin of the $E^t_{S_1}$ barrier in NeoR. a** Representation of the increase in the distance between the negative and positive charge centroids due to the positive charge confinement along the C13=C14 and C9=C10 photoisomerization paths. Comparison between electrostatic potential (ESP) maps indicates that along the C9=C10 coordinate the extent of the confinement is less pronounced being the charge at the CoIn spread on a longer rPSB chromophore moiety (i.e., C10-C11-C12-C13-C14-C15-N). **b** Proposed origin of the isomerization barrier in terms of COV energy ($H_{COV}$, dashed line) destabilization due to the charge confinement resulting from the mixed [CT] · c'[COV] to the pure [COV] electronic structure change along the $S_1$ adiabatic energy profile (in orange). We hypothesize that the diabatic energy curves cross halfway along the isomerization coordinate $\alpha$, which therefore corresponds to the point with the highest diabatic coupling. The such diabatic coupling will then vanish at the CoIn. The left and right panels display the shape of the $S_1$ and $S_0$ adiabatic potential energy curves along the BLA$_{PSB}$ coordinate (see definition in the caption of Fig. 4) at FS (left) and CoIn (right) and are in line with the presented FC→FS and CoIn computations. The BLA$_{PSB}$ and $\alpha$ coordinates are substantially orthogonal.

base moiety (see the resonance formula in Fig. 3a). The highly delocalized electronic structure of the FS points to close $H_{COV}$ and $H_{CT}$ values consistent with the small Charge$_{PSB}$ value of $S_1$. On the other hand, the large $S_1$ Charge$_{PSB}$ computed in the CoIn region points to an adiabatic state dominated by the COV diabatic. Thus, the charge confinement effect described above is translated into an increase in the weight of the COV diabatic state along the isomerization coordinate. This justifies the steep $S_1$ energy increase in the CoIn region that would originate from the simultaneous increase in the COV weight and COV destabilization due to the offset in the electrostatic interaction with E141.

Such a delocalization-confinement mechanism (i.e., without an initially delocalized charge there is no progressive confinement) also suggests a lesser sensitivity of the $S_0$ state to the progression along the isomerization coordinate. In fact, while $S_0$ becomes progressively dominated by the CT diabatic state, in CT the rPSB charge remains relatively unconfined along a long segment (-C7-C8-C9-C10-C11-C12-C13-) of the rPSB backbone even in the CoIn region.

Above we have shown that, the conventional C13=C14 isomerization preventing microbial rhodopsin to be highly fluorescent, can be blocked or slowed down by a suitable change in the electrostatic environment of the all-*trans* rPSB chromophore. More specifically, the presented *a*-ARM$_{E141}$ model, indicates that NeoR is the product of an evolutionary process driven by the translocation of a negatively charged residue from the chromophore Schiff base region

to a region located halfway along the chromophore conjugated chain (E141). The main local effect of this process is the generation of a DA state featuring a delocalized rPSB-positive charge. The rest of the described properties, including spectral properties such as the large bathochromic shift, small Stokes shift, and sizable $S_1$ isomerization energy barrier along the canonical C13=C14 torsional coordinate, are a consequence of such a change. This behavior was recently documented by El-Tahawy et al. in isolated rPSB chromophores subject to homogeneous, strongly negative red-shifting electric fields and, therefore, the "two electron-two orbital model" theory that is shown in there to account for the electrostatic origin of the C13=C14 photoisomerization energy barrier appears also operative in the here presented NeoR[2]. Notice that the described E141 counterion location is presently a theoretical result that remains to be experimentally demonstrated.

The effect associated with the repositioning of the rPSB counterion was previously proposed to explain the $\lambda^a_{max}$ changes observed in a set of rhodopsin mimics based on the human cellular retinol-binding protein II (hCRBPII)[42]. Similar to our NeoR model, the members of the set displaying a large red shift were found to be associated with a counterion located far from the Schiff base moiety. This conclusion was reached through X-ray crystallographic analysis supporting the hypothesis that a red-shifted $\lambda^a_{max}$ must be associated with an even distribution of the iminium charge along the chromophore $\pi$-conjugated chain. Such delocalization can be associated with the rPSB

delocalized charge seen in the $a$-ARM$_{E141}$ calculation and interpreted as a COV$\longleftrightarrow$CT resonance hybrid (see Fig. 3a) or, in a different language, to a near cyanine limit situation[43].

$a$-ARM$_{E141}$ leads to a possible general principle for the engineering of other highly fluorescent rhodopsins that we call "delocalization-confinement". Such principle establishes that the electrostatic field generated by the cavity, for instance via a specific counterion localization, must yield a vastly delocalized geometrical and electronic structure of the rPSB conjugated chain in both the DA and the FS state of the protein. In this condition, an electrostatically induced high reaction barrier can be generated via a rPSB charge confinement process occurring, unavoidably, along the canonical C13=C14 isomerization path in the region entering the corresponding CoIn channel.

Finally, the reported results provide evidence that the spectroscopy of retinal proteins is regulated by the same principles regulating GFP-like fluorescence. More specifically, it was proposed that the GFP variants achieving maximal $\pi$-electron delocalization (called the cyanine limit) are the ones where a COV and CT configurations of the protein chromophore have exactly the same weight, thus pushing the $\lambda^a_{max}$ value to the extreme red and culminating in a null Stokes shift.

## Methods

The employed hybrid QM/MM modeling of NeoR was performed using the $a$-ARM protocol[20–22] and based on the comparative model structure built and validated by S. Adam et al.[7]. Further details about the $a$-ARM protocol are given as Supplementary Informations (Supplementary Section 1). After initially producing, automatically, default $a$-ARM models for WT and mutant NeoR, the equilibrium geometries of the DA were obtained via re-assignment of the counterion before carrying out ground state geometrical relaxation with energy gradients calculated at the 2 root state average CASSCF(12,12)/6-31 G*/AMBER94 level of theory[44–46] using the Molcas/Tinker[47,48] interface (Supplementary section 2). The relevant energies were instead computed, again employing the Mocas/Tinker interface, at the single-state and, in specified cases, multistate multiconfigurational levels. These correspond to the 3-root state average CASPT2(12,12)/6-31 G*/AMBER94 and 3-root state average XMCQDPT2/CASSCF(12,12)/6-31 G*/AMBER94 levels, respectively. The XMCQDPT2 calculation was based on Firefly v8.2[49]. The collection of geometries connecting the FS to the different CoIn's and defining the MEPs discussed above, were obtained via constrained geometry optimization at the 2 root state average CASSCF(12,12)/6-31 G*/AMBER94 level of theory. As shown in Supplementary Section 5, for each MEP the $S_1$ isomerization barrier $E^f_{S1}$ is estimated after revaluating the $S_1$ energy profiles at the above CASPT2 and XMCQDPT2 levels via single point energy calculations and measuring the energy difference between the highest $S_1$ energy value (i.e., corresponds to the CoIn) and the $S_1$ energy value of the FS.

## Data availability

The cartesian coordinates of the DA and the FS of the QM/MM model of NeoR generated in this study are provided, respectively, in Supplementary Data 1 and Supplementary Data 2. Source data are provided with this paper.

## Code availability

As stated in the Method section, all the calculations carried out in this work were performed using a combination of the quantum chemical program MOLCAS and molecular mechanics program TINKER except for XMCQDPT2 calculations which were performed using the computational chemistry program Firefly.

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

## Acknowledgements
The research has been in part supported by Grants NSF CHE-CLP-1710191 and NIH GM126627-01. The authors are also grateful for a Department of Excellence Grant 2018-2022 funded by the Italian MIUR and by Fondazione Banca d'Italia for providing equipment funds. D.P. acknowledges the Italian Ministry of University and Research (MUR) for a Rita Levi Montalcini grant. We thank Suliman Adam for having provided the NeoR homology model used in this work. I.S. acknowledges support by the DFG through SFB 1078, project C6. I.S. thanks the Israel Science Foundation (Research Center grant no. 3131/20). L.B. and M.O. acknowledge partial support from European Union, Next Generation EU, MUR Italia Domani Progetto mRNA Spoke 6 del "National Center for Gene Therapy and Drugs based on RNA Technology", CUP di progetto B63C22000610006.

## Author contributions
M.O. and R.P. conceived and designed the work. R.P. generated the QM/MM models and carried out all the calculations. All the authors R.P., L.B., L.P.G., D.P., I.S., and M.O. contributed to analyzing, discussing, and interpreting the results of the calculations. M.O. and R.P. wrote the manuscript.

## Competing interests
The authors declare no competing interests.
