## [Peer Review File · Nature Communications]

REVIEWER COMMENTS

Reviewer #1 (Remarks to the Author):

Neorhodopsin (NeoR) is a recently (2020) discovered archeal retinal protein presenting unexpected extreme photophysics: a reddish 690nm absorption (displaying the strongest bathochromic shift among all known microbial rhodopsins) and an unexpected and surprisingly highly fluorescent (FQY=20%) long excited state lifetime that is just slightly more red-shifted (707nm) (at odd with other retinal proteins - either animal or archeal - that are known to only very weakly fluoresce (FQY<10⁻⁴) owing to ultrafast and competitive photoisomerization non-radiative decay channels of their retinalPSB chromophore).

NeoR is here scrutinized by static QM/MM computations aiming at replicating its spectroscopic features and explaining its high FQY. A model for NeoR (that lack a reference experimental (e.g., crystallographic) structure) is thus designed employing an automatic protocol (developed and previously validated by the same authors) and based on a comparative model found in the literature. CASSCF/MM geometry optimizations of the relevant states and photoisomerization paths (complemented by single point PT2/MM energy corrections) are eventually performed to track the photoisomerization channels and their energy profiles and replicate spectroscopic features.

Deprotonated 141 (the residue laying closer to the beta-ionone ring among the nearby carboxylate residues possibly serving as counterions) is suggested to be the retinalPSB counterion, thus creating the electrostatic environment responsible for NeoR "strangeness". Specifically, an electrostatic model, complemented with ground and first excited state charge&wavefunction analysis and its interpretation in terms of strong mixing between diabatic CT/COV states (leading to poor, if not negligible, photoinduced CT and a positive charge that is spread all over the polyene chain on both the ground and excited states) is presented to account for the observed behavior, namely the appearance of significant energy barriers along the photoisomerization paths that prevent ultrafast internal conversion and fluorescence quenching. Calculations on mutants are also presented to support the model and protonation states of nearby titrable residues.

While this is certainly an interesting problem, I have major concerns impacting both the solidity and novelty of the conclusions, as well as the completeness of the bibliography, that prevent by now publication on NatureComm, namely:

1) An extensive model calling for electrostatic effects in dictating the photophysical properties of retinalPSB chromophores has been already presented in the past (see JCTC 2016, 12, 4460). The

authors seem to ignore these findings that, among the many other scenarios illustrated there, did show exactly the same case reported here (i.e., electrostatic effects leading to red-shift and increased fluorescence). This work replicates the old results, including the wavefunction/charges analysis. In particular, the strong mixing between CT/COV diabatic states (that we can use as an electronic basis-set to represent the eigenfunctions of S1 and S0), as well as their avoided or real crossings, were already seen to dictate electronic barriers and photophysical properties of retinal chromophores with strongly red-shifted absorptions (as it is the case for NeoR). These include bond lengths equalization at the excited and ground state minima (DA and FS in the present work), as well as disfavoring photoisomerization internal-conversion decay paths and switching between a nonfluorescent ultrafast photochromism (i.e., photoisomerization) into a fluorescent state upon the effects of external electric fields generated either by external charges or exogenous fields inducing an absorption red-shift.

The authors should fairly cite this work and its follow up paper (ANGEW 2020, 59, 20619), and put their findings in the context of the generalized model presented there and its predictions for the spectral and photochemical tuning of retinal chromophores and mimics. The results should be also put in the context of the two-electrons/two-orbitals model that is shown there to explain the photoinduced behavior in strongly polar conjugated polyenes when strongly negative red-shifting electric fields are used.

In conclusion, neither the presented results, nor their interpretation, are surprising as they do faithfully replicate what shown already, the interesting thing just relying on a real system (NeoR) that was unknown in 2016 and that supports, together with the present theoretical results, the prediction of a fluorescent state upon specific red-shifting electrostatic conditions.

2) Despite what some people may think, I share the authors' view that static non-dynamical descriptions based on surface mapping and geometry optimizations (that this work is based on) can still be informative and work reasonably well when an handful of useful (although mainly qualitative) information have to be delivered and an interpretative model has to be drawn. However, for this to happen, a systematic study is required. The presented work and data lack this completeness. Indeed, bond order analysis (Table S4), as well as the chromophore structure displayed at the two (DA and FS) minima, shows very equalized bond lengths, mainly towards the CN side of the chromophore, thus making hard to decide which C-C bond should be twisted in searching for favored isomerizations. To consolidate the claim that a stable minimum exists on S1 and is responsible for NeoR fluorescent, the authors should also scan rotations along "single" bonds (C8-C9, C10-C11, C12-C13), as well as optimize corresponding CoIns. Otherwise, favored isomerization paths on S1, leading to internal conversion and fluorescence quenching, cannot be excluded.

Again, the extensive model already presented in the literature shows that electrostatic conditions triggering red-shifted absorption/emission, while creating CT/COV avoided crossing barrier towards

double bond twisting, may actually result into rather low energy barriers and conical intersections for rotation around C10-C11 and C12-C13 "single" bonds. The work should thus be integrated with further analysis to prove (at least qualitatively and based on static mapping of S1) that the S1 minimum is stable enough to account for the long excited state lifetime and strong fluorescence observed.

By now, the work is incomplete and the data reported do not suffice in supporting the claims.

3) While the model with no counterion (i.e., all four residues are protonated) produces an absorption closer to the experiment, it is discarded as it is claimed to lack a stable FS structure. However, the authors do not report the data supporting this claim. Additionally, FIG4 shows not negligible energy barriers along all the mapped photoisomerization channels with charges switched off (grey and yellow squares): since switching off counterion charges should replicate the model with no charged counterions, this seems to contradict the claim and should be explained. Finally, these results would also support what one of the authors already reported in the past (JACS 2004, 126, 16018): a counterion placed above the retinal plane (such as 141) would lock the photoisomerization paths of the nearby double bonds below, thus effectively acting as a control knob. This is also overlooked and should be properly referred to.

4) Eventually, it would be great to show what happens when different electrostatic environments (pushing absorption progressively towards the blue and/or when 141 is protonated) are experienced by the retinal chromophore: e.g., D140 or E132 (that replicate bR absorption) and E136 E262 or D140 E262 (that absorb as in solvent). For the authors' claim to be solid, the system should not be fluorescent anymore.

Additional minor remarks follow:

Data for the first two points (FC and FS) of the 4 displayed MEPs in the 4 top panels of FIG4 should be the same: however it does not look like that. Correct or explain.

The shoulder observed in the experimental absorption band is not reproduced and it is assigned to a vibrational feature. However, it may be also explained by equilibrium with a different protomer absorbing at more blue shifted energies. Please, comment.

Fig3c: is the ground state wf representation for the FS structure correct? I would have expected COV +c'CT instead. Fig5b (central panel) seems to support this view.

Ref 2 lacks pages.

Reviewer #2 (Remarks to the Author):

This work investigates the unusually high fluorescence yield of neorhodopsin, a newly discovered opsin protein, by computational modelling. A comparative model of neorhodopsin is used and further refined by combined quantum mechanical/molecular mechanical computations to obtain information on the protonation states of residues near the retinal Schiff-base. These have notable influence on the absorption and emission properties of the retinal chromophore. The authors identify a charged glutamate approximately in the middle of the chromophore (rather than at its typical position at the Schiff base end) which acts as a counterion and reproduces the experimental absorption and emission values. This residue is also responsible for confining the electronic delocalization of retinal in the excited state, which creates significantly high barriers among the double bonds that prevent photoisomerization and thus keeps the molecule in a fluorescent state.

This is a solid computational study and the authors quantify their findings by numerous computations deciphering the effect of the protein environment and the special charged residue on the electronic nature of the chromophore. The methods used here are state-of-the-art, most relevant work is cited. The supporting information includes detailed instructions to reproduce the computations. The interpretations are sound, although not always easy to follow (see below). Similar aspects were also discussed in previous work (e.g. JACS 2004, 126, 49, 16018–16037), albeit there focusing on bond selectivity and not mentioning the here presented concept of delocalization confinement. The authors state that experimental evidence for the location of the counterion still remains to be demonstrated through experiments.

Although this is, as mentioned, a solid computational work, I am not confident that it meets the criteria for urgent publication and is suitable for the broad audience represented through Nature Comm. I would rather recommend to publish it in a more specialized journal, like e.g. J. Chem. Inf. Model., after some minor issues have been clarified by the authors:

1) The explanation of the effects in this work appears somewhat abstract. E.g. it is not easy to follow the argumentation on the changes in state character (COV vs. CT) as well as to decipher the notation [CT]-c[COV] vs. [CT]+c'[COV] in Fig. 3.

2) The authors discuss an unusually large CT weight in the ground state. This discussion is purely based on Mulliken charge analyses. Can this and also the state characters in S1 be better quantified e.g. through analysis of the state coefficients in (XMS)-CASPT2?

3) The authors base the discussion and investigations on the clockwise (CW) torsional motion along C13=C14 and therefore assign the motions of the remaining double bonds accordingly. Other proteins (e.g. Channelrhodopsin) give rise to the existence of both, CW and anti-CW paths, why the anti-clockwise channel has been disregarded in this model?

Reviewer #3 (Remarks to the Author):

The authors present a computational study of the photophysical properties of the recently discovered Neorhodopsin. The whole protein environment is taken into account, and the peculiar properties of NeoR are explained in terms of environmental effects on the delocalized charge distribution of the chromophore.

The work is sound and well written and deserves to be published.

Here are my remarks

1) In the abstract, the author say that the chromophore considered features "a highly diffuse charge density". Although this will become clearer in the manuscript, this does not seem to be a very remarkable feature for a conjugated chain.

2) The authors claim that NeoR features a near-infrared absorption band (690 nm). As far as I know, the IR should start at 780-750 nm, so 690 is still Vis.

3) Very large differences in the vertical excitation energies are found by changing the protonation sites (figure 2B). Are these differences originated by a geometrical effect or

by different state-specific response of the QM subsystem (or a combination of the two effects)? In other words, the large differences in the vertical excitation energies are still present if a common geometrical arrangement is used for the different protonation states considered?

4) In figure 2B, what is the meaning of the sentence "This model does not exhibit a DA"? The chromophore dissociates or isomerizes?

5) In figure 3C, the two wavefunctions $[CT]+c'[COV]$ and $[CT]-c'[COV]$ are not orthogonal. Maybe the ground state should be $[COV]+c'[CT]$?

6) If the energies of the two diabatic states COV and CT are close, the two states experience a large diabatic coupling at the DA geometry (about 0.8 eV, i.e., half of the S1-S0 energy difference). This very large diabatic coupling should decrease along the isomerization coordinate, vanishing at the CoIn. The authors should comment in this respect.

7) Figure S8: according to the caption, the map has been generated using the coordinates X1 and X2, but other coordinates are shown in the figure. Do X1 and X2 span the same space of BLApsb and alpha?

Some typos:

- line 201: erase "more"

- line 256: "qualitative" should read "qualitatively"

- line 390: charge -> charged

The reply to the reviewers is structured in the following way: reviewer comments (in black), author comments/reply (in blue), author proposed changes (in red).

REPLY TO THE REVIEWERS

Reviewer #1 (Remarks to the Author):

"...While this is certainly an interesting problem,..."

We are grateful to the reviewer for his/her accurate summary and for the generally positive evaluation of our work.

I have major concerns impacting both the solidity and novelty of the conclusions, as well as the completeness of the bibliography, that prevent by now publication on NatureComm, namely:

(1) An extensive model calling for electrostatic effects in dictating the photophysical properties of retinal PSB chromophores has been already presented in the past (see JCTC 2016, 12, 4460). The authors seem to ignore these findings that, among the many other scenarios illustrated there, did show exactly the same case reported here (i.e., electrostatic effects leading to red-shift and increased fluorescence). This work replicates the old results, including the wavefunction/charges analysis. In particular, the strong mixing between CT/COV diabatic states (that we can use as an electronic basis-set to represent the eigenfunctions of S1 and S0), as well as their avoided or real crossings, were already seen to dictate electronic barriers and photophysical properties of retinal chromophores with strongly red-shifted absorptions (as it is the case for NeoR). These include bond lengths equalization at the excited and ground state minima (DA and FS in the present work), as well as disfavoring photoisomerization internal-conversion decay paths and switching between a nonfluorescent ultrafast photochromism (i.e., photoisomerization) into a fluorescent state upon the effects of external electric fields generated either by external charges or exogenous fields inducing an absorption red-shift.

The articles pointed out by the reviewer have now been cited in the main text of the revised manuscript. We feel sincerely indebted to the reviewer for making us aware that the described relationship between red-shifted absorption and increase in the C13=C14 barrier had been predicted. On the other hand, we reckoned that the authors of the same articles underline that (JCTC 2016, 12, 4460):

“We could speculate that the encountered behavior in a strong negative electric field accounts for the lack of retinal proteins absorbing into the red above 590 nm, as this is exactly the absorption energy window caused by an electric field of -0.0025 to -0.0030 au which leads to bond length equilibration in the ES ...”.

The above sentence reinforced our believe that understanding how exactly NeoR is capable to bring about such previously unexpected absorption maximum is of fundamental importance. We show that this is due to the details of the complex "biological electrostatics" generated by the specific NeoR protein environment not considered in the above references. In the revised

main text we now mention that: (1) our target is not formulating a general model for the effects of electrostatics on the retinal chromophore, but to provide a *specific* explanation valid in the case of NeoR and thus (2) we focus on the field generated by the NeoR cavity along its specific isomerization coordinate. Since the article mentioned by the reviewer consider, appropriately in that context, a model using a uniform electrostatic field acting on an isolated retinal chromophore, our model using the complex electrostatic field generated by the protein cavity demonstrates that the protein environment can lead to different results.

CHANGES:

Main manuscript, lines 48-56 in the Introduction:

“Deciphering how natural evolution in NeoR has tuned these extreme absorption and emission properties of the rPSB chromophore could expand our ability to design optogenetic tools with augmented functionality. On this regard, few studies¹⁰⁻¹² have formulated rules for tailoring the retinal chromophore properties based on the effects of uniform electrostatic field acting on isolated chromophores or via chromophore chemical modifications. Here, we focus on the effect of the complex of protein environment not explicitly consider in the above studies. Therefore, the modeling of NeoR represents a new learning-opportunity that can be also used to assess the transferability of the rules mentioned above.”

and lines 422-426 in the Conclusions:

“This behavior was recently documented by El-Tahawy et al. in isolated rPSB chromophores subject to an homogeneous, strongly negative red-shifting electric fields and, therefore, the “two electron-two orbital model” theory that is shown in there to account for the electrostatic origin of the C13=C14 photoisomerization energy barrier appears also operative in the here presented NeoR¹¹.”

In conclusion, neither the presented results, nor their interpretation, are surprising as they do faithfully replicate what shown already, the interesting thing just relying on a real system (NeoR) that was unknown in 2016 and that supports, together with the present theoretical results, the prediction of a fluorescent state upon specific red-shifting electrostatic conditions.

We respectfully disagree with the reviewer. While it is correct to claim, that the general aspects of the effects of cavity point charges have been discussed in the past, the reviewer has to admit that the mechanism for the NeoR fluorescence we propose is original (Reviewer #2 agrees). This is also stressed in the title of the paper that explicitly refer to a "sequential charge delocalization and re-localization mechanism".

2) Despite what some people may think, I share the authors' view that static non-dynamical descriptions based on surface mapping and geometry optimizations (that this work is based on) can still be informative and work reasonably well when an handful of useful (although mainly qualitative) information have to be delivered and an interpretative model has to be drawn. However, for this to happen, a systematic study is required. The presented work and data lack this completeness. Indeed, bond order analysis (Table S4), as well as the chromophore structure displayed at the two (DA and FS) minima, shows very equalized bond lengths, mainly towards the CN side of the chromophore, thus making hard to decide which C-C bond should be twisted

in searching for favored isomerizations. To consolidate the claim that a stable minimum exists on S1 and is responsible for NeoR fluorescent, the authors should also scan rotations along "single" bonds (C8-C9, C10-C11, C12-C13), as well as optimize corresponding CoIns. Otherwise, favored isomerization paths on S1, leading to internal conversion and fluorescence quenching, cannot be excluded.

In order to satisfy the reviewer requests, we now have:

- (1) computed the relaxed scans along each double bond (Fig. 4 and S5)
- (2) computed the relaxed scans along each single bond (Fig. S9 and S10 below).
- (3) located all corresponding conical intersections.

We find that, in all cases, the barrier is > 15 kcal/mol.

All additional results are now mentioned in the revised main text and reported in the revised supporting information. With absolute certainty we find that the modelled NeoR fluorescent state is highly stable as already indicated by the inspection of Fig. 3D reporting on the bond-length-alternation values at such a point.

Figure S9. Respectively from left to right: photoisomerization MEP associated with the CW rotations of C12-C13, C10-C11, C8-C9 rPSB single bonds. Energies are relative to the DA. For each MEPs, 3 root state average CASPT2/CASSCF(12,12)/6-31G*/AMBER94 (CASPT2/AMBER) energy profiles are reported. $E_{S_1}^f$ indicates the S_1 isomerization barrier and corresponds to the CoIn and FS S_1 energy difference.

Figure S10. Respectively from left to right: photoisomerization MEP associated with the CCW rotations C12-C13, C10-C11, C8-C9 rPSB single bonds. Energies are relative to the DA. For each MEPs, 3 root state average CASPT2/CASSCF(12,12)/6-31G*/AMBER94 (CASPT2/AMBER) energy profiles are reported. $E_{S_1}^f$ indicates the S_1 isomerization barrier and corresponds to the CoIn and FS S_1 energy difference.

CHANGES:

Revised main text, lines 287-290 in the Results and Discussion section:

”In order to check the existence of alternative and lower energy photoisomerization channels, we computed the C8-C9, C10-C11 and C12-C13 single bond MEPs and found that in all cases $E_{S_1}^f$ is greater than 20 kcal/mol⁻¹ (see Fig S9 and S10). This supports the high stability of the FS displayed by our a-ARME_{E141} model.”

We inserted Fig. S9 and S10 and the corresponding discussion in the revised supporting information, lines 394-401 in section S5:

“In addition to the MEPs along the rPSB double bonds we computed (see Fig. S9 and S10) the C8-C9, C10-C11 and C12-C13 single bond MEPs to make sure that there are energy barriers preventing the excited state isomerization along these coordinates. The resulting 21, 21, 41 and 23, 20, 33 kcal mol⁻¹ CASPT2 energy barriers for CW and CCW rotations respectively support the stability of the FS of our a-ARME_{E141} model. In conclusion, we found that the S_1 rotation of each chromophore bond comprised between C7 and C14 atoms of the NeoR is associated with a barrier greater than 15 kcal/mol⁻¹ (Fig. S5, S9 and S10).”

3) While the model with no counterion (i.e., all four residues are protonated) produces an absorption closer to the experiment, it is discarded as it is claimed to lack a stable FS structure. However, the authors do not report the data supporting this claim.

The data have now been added to the Supporting Information section. In short, we found a negligible barrier when following the same protocol used to investigate the selected NeoR model.

Additionally, FIG4 shows not negligible energy barriers along all the mapped photoisomerization channels with charges switched off (grey and yellow squares): since switching off counterion charges should replicate the model with no charged counterions, this seems to contradict the claim and should be explained.

As shown in Fig. 4 the S_1 energy profiles recomputed after switching off the charges of the E141 counterion of a-ARME_{E141} yields a strong reduction of the C13=C14 barrier even when the geometries defining the path are not re-optimized. Therefore, this behavior appears in line with the lack of a stable FS structure in the model with no counterions since switching off the charges of E141 should roughly replicate the electrostatic embedding imposed by that model. Notice that, this effect of barrier reduction decreases along the series and disappears at C7=C8, that however displays a large steric effect.

A better explanation has been added to the main text.

Finally, these results would also support what one of the authors already reported in the past (JACS 2004, 126, 16018): a counterion placed above the retinal plane (such as 141) would lock the photoisomerization paths of the nearby double bonds below, thus effectively acting as a control knob. This is also overlooked and should be properly referred to.

We agree with the reviewer and, now, cite the related reference. However, again, our target is to model a specific but real system: NeoR. For this reason, we have tried to avoid (in this specific context) the use of simple models such as the isolated counterion - chromophore complex. This has now been further stressed.

On the other hand, to be accurate, the selectivity rules formulated in that reference are not fully consistent with the reviewer remark. In fact, it is claimed:

“... Isolated systems (1 and 2) or Central models (1a, 2a, and 2b) display the most favored conditions for ultrafast and efficient central double-bond photoisomerizations ...”. The indexes are related to the following figure:

now this seems exactly opposite to the reviewer claim:

“a counterion placed above the retinal plane (such as 141) would lock the photoisomerization paths of the nearby double bonds below, thus effectively acting as a control knob”.

We therefore feel that what is claimed in our paper, could at least clarify what the effect actually is in the "biological" chromophore environment of NeoR.

However, we think that the citation above is appropriate when discussing the fact that in NeoR the favored isomerization paths are the one relative to the C11=C12 e C9=C10, namely the one closer to the counterion. Indeed, the reference report that:

“Moving the counterion above the molecular plane of the chromophore does effect competitive isomerizations for internal C-C double bonds (see 2a and 2b), opening or locking specific isomerization paths: a barrierless (i.e., efficient) photoisomerization leading to a TICT CI point (i.e., an ultrafast radiationless decay funnel) occurs only for the double bond being closer to the anion (i.e., the double bond right below it) ...”

As already pointed out however, the transferability of those results is not straightforward given the complexity of the NeoR electrostatic field.

CHANGES:

We commented on the absence of a stable FS displayed by the model without counterions in the revised supporting information, lines 97-101 in section S2:

“As shown in Fig. 2B we found that the model with no counterions (i.e. with all E136, D140, E141 and E262 residues protonated) does not exhibit a stable S_1 minimum and therefore, given the absence of a minimum on S_1 , the energy gradient controlling the S_1 optimization steered the rPSB directly to the CoIn that was found associated to a nearly 90° twisted conformation along the C12-C13 bond.”

We commented on the consistency between the S_1 energy profiles reported in Fig. 4 and the lack of a stable S_1 minimum displayed by the model with no counterion in the revised main text, lines 318-321 in the Results and Discussion section:

“This behavior is consistent with the lack of a stable FS (i.e. due to the absence of a S_1 energy barrier controlling access to the CoIn) displayed by the model with no counterion (see Fig. 2B). In fact, switching off the charge of the E141 counterion roughly replicates the electrostatic embedding imposed by that model.”

We added the citation of the ref. JACS 2004, 126, 16018 concerning the control of the isomerization regioselectivity in the revised main text, lines 278-283 in the Results and Discussion section:

“This result appears consistent to what was found in the past in a work of Cembran et al. aimed at investigating the relationship between the position of an acetate counterion and the photoisomerization of a nearly isolated protonated polyene chain¹⁰. They found that placing the counterion above the polyene favors the isomerization of the double bonds closest to the counterion; this result can be loosely associated to the favored C11=C12 e C9=C10 isomerization in our a-ARME141 model.”

4) Eventually, it would be great to show what happens when different electrostatic environments (pushing absorption progressively towards the blue and/or when 141 is protonated) are experienced by the retinal chromophore: e.g., D140 or E132 (that replicate bR absorption) and E136 E262 or D140 E262 (that absorb as in solvent). For the authors' claim to be solid, the system should not be fluorescent anymore.

As shown by our additional computations whose result is reported Fig. S11 (see below), all S_1 isomerization paths display a flat and substantially barrierless energy profile (orange profile). This is not consistent with the observed excited state lifetime of NeoR estimated to be 1.1 ns. Therefore, the presence of the counterion in position E141 appears to be critical for the generation of a barrier. The models with no counterion, or the counterion in positions E136 or E262 are therefore unrealistic.

Figure S11. Respectively from left to right: photoisomerization MEP associated with the CW rotation of the C13=C14 double bond for NeoR models featuring E262 (a-ARME262), D140 (a-ARMD140) and E136 (a-ARME136) residues as rPSB counterions. S_0 (in blue), S_1 (in orange) and S_2 (in green) energies are relative to the DA. For each MEPs, 3 root state average CASPT2/CASSCF(12,12)/6-31G*/AMBER94 (CASPT2/AMBER) energy profiles are reported.

CHANGES:

We commented on this regard in the revised main text, lines 322-327 in the Results and Discussion section:

”The models featuring a counterion configuration different from that of a-ARME141, display a flat, and substantially barrierless, S_1 isomerization energy profiles (see Fig. S11). As stated above, this is not consistent with the excited state lifetime of NeoR estimated to be 1.1 ns. Therefore, our data indicate that a E141 counterion appears not only critical for tuning the extreme spectroscopy of NeoR (see Fig. 2B) but also for the generation of a barrier.”

And reported the corresponding data in the revised supporting information (see Fig. S11).

Additional minor remarks follow:

Data for the first two points (FC and FS) of the 4 displayed MEPs in the 4 top panels of FIG4 should be the same: however it does not look like that. Correct or explain.

We are grateful to the reviewer for pointing this out. After checking, we detected a typo in the figures that has now been corrected. Accordingly, we have modified Fig. 4 and related figures (Fig. S6-S8) in the Supporting Information.

CHANGES:

We included modified Fig. 4 in the revised main text and Fig. S6 and S7 in the revised supporting information.

The shoulder observed in the experimental absorption band is not reproduced and it is assigned to a vibrational feature. However, it may be also explained by equilibrium with a different protomer absorbing at more blue shifted energies. Please, comment.

While an in-depth computational investigation of the origin of such shoulder goes beyond the scope of the present work, the experimental spectrum (see below) suggests a vibronic origin of the shoulder. More specifically, the absorption band can be seen composed by one main intense band attributable to the transition from the lowest vibrational state in S_0 to the lowest vibrational state in S_1 and by one shoulder attributable to the transition to a higher vibrational state in S_1 . The specular shape of the emission band can be interpreted in a similar fashion, with radiative decay from the S_1 ground vibrational state to a couple of S_0 vibrational states. In contrast, a shoulder originating from a different protomer would likely (assuming similar Stokes shifts for both protomers) yield a fluorescence spectrum with maxima in the same order as they appear in the absorption spectrum. Of course, one could get different Stokes shifts for distinct protomers if one of them has, for instance, a largely displaced S_1 geometry with respect to the S_0 geometry. However, such a geometrical displacement would likely occur along torsional coordinates, and this is not consistent with the FS stability. Moreover, from the low Stokes shift we can deduce that S_0 and S_1 minima have very similar geometries, and that the displacement can be mainly attributed to the BLA (vibronic peak distance $\sim 1400 \text{ cm}^{-1}$). This conclusion is in line with (i) a single protomer picture (ii) all other computational observations on the key structures.

Fig3c: is the ground state wf representation for the FS structure correct? I would have expected COV +c'CT instead. Fig5b (central panel) seems to support this view.

Thank you for raising this question. We have now corrected the notation in Fig. 3C.

CHANGES:

We reported the corrected Fig. 3C in the revised main text.

Ref 2 lacks pages.

This has now been corrected. Thank you for the careful reading.

Reviewer #2 (Remarks to the Author):

"...This is a solid computational study and the authors quantify their findings by numerous computations deciphering the effect of the protein environment and the special charged residue on the electronic nature of the chromophore. The methods used here are state-of-the-art, most relevant work is cited. The supporting information includes detailed instructions to reproduce the computations. The interpretations are sound, although not always easy to follow (see below). Similar aspects were also discussed in previous work (e.g. JACS 2004, 126, 49, 16018–16037), albeit there focusing on bond selectivity and not mentioning the here presented concept of delocalization confinement. The authors state that experimental evidence for the location of the counterion still remains to be demonstrated through experiments...."

We are grateful to the reviewer for his/her accurate summary and for the generally positive evaluation of our work.

"...I am not confident that it meets the criteria for urgent publication and is suitable for the broad audience represented through Nature Comm. I would rather recommend to publish it in a more specialized journal, like e.g. J. Chem. Inf. Model., after some minor issues have been clarified by the authors..."

We respectfully disagree with the reviewer. NeoR has been puzzling the photochemist and photobiologist communities since its discovery. As reported in the literature and in the cited theoretical work (see Reviewer #1 suggested reference), the existence of such a rhodopsin was believed to be impossible. By demonstrating a novel "delocalization and confinement" mechanism we have explained, at the atomic and electronic level, the intense NeoR emission. Clearly, this not only provides a novel fluorescence mechanism, but also new "rules" useful for the engineering of highly fluorescent rhodopsin starting from non-fluorescent ones.

1) The explanation of the effects in this work appears somewhat abstract. E.g. it is not easy to follow the argumentation on the changes in state character (COV vs. CT) as well as to decipher the notation $[CT]-c[COV]$ vs. $[CT]+c[COV]$ in Fig. 3.

Our explanation is rooted in a valence-bond picture of the S_1 electronic structure of NeoR. More specifically, the electronic structure is represented by the mixing of two "resonance" structures characterized by different charge distributions: the charge-transfer (CT) and covalent/diradical (COV) structures. The $[CT]-c[COV]$ electronic structure notation, simply informs on the above fact. Our calculations show that the relative weights (i.e. the c coefficient) of these resonance structures changes along the isomerization path passing from a CT dominated situation near the FS state to a fully COV dominated situation near the conical intersection. Thus, when following basic state-mixing ideas, the barrier originates from the mixing of the COV and CT components to generate the adiabatic S_0 and S_1 states which is maximized when they have the same energy. Accordingly, in our model the CT and COV states as well as the periodic function describing their coupling are obtained by imposing that the computed adiabatic S_1 and S_0 energy profiles are the eigenvalues of a model Hamiltonian with the CT and COV energies (called diabatic energies) along the diagonal and their mixing as the off-diagonal element.

The Hamiltonian above is a commonly adopted model in photochemical mechanistic studies. Such model is strongly supported by the charge distribution evolution calculated along the S_1 reaction path that shows a clear charge migration when moving from the CT-like charge distribution dominating the FS region to the conical intersection region where a COV-like charge distribution dominates. Such motion is clearly illustrated in Fig. 4.

2) The authors discuss an unusually large CT weight in the ground state. This discussion is purely based on Mulliken charge analyses. Can this and also the state characters in S_1 be better quantified e.g. through analysis of the state coefficients in (XMS)-CASPT2?

We have tested the stability of the description based on Mulliken charges in the past comparing the values obtained at different level of theory; typically, CASSCF, CASPT2 and XMC-QDPT2 (similar to XMS-CASPT2). On the other hand, we usually avoid comparing the coefficients because, obviously, they depend on the orbital shape and order which may change in different calculations (the total energy is an invariant with respect to the mixing of the active space orbitals). A solid interpretation based on coefficients would therefore require an orbital localization. To avoid that and the related we decided to rely on the charge

distribution "index" calculated in the present paper.

3) The authors base the discussion and investigations on the clockwise (CW) torsional motion along C13=C14 and therefore assign the motions of the remaining double bonds accordingly. Other proteins (e.g. Channelrhodopsin) give rise to the existence of both, CW and anti-CW paths, why the anti-clockwise channel has been disregarded in this model?

The choice of the CW rotation of C13=C14 is based on the direction of the double bond pre-twisting found in both the dark-adapted state (DA) and the fluorescent state (FS) of the presented NeoR model. It is commonly observed that the direction of the pre-twisting of the reactive double bonds, imposed by the steric and electrostatic effects of the protein cavity, biases the isomerization of the chromophore in the CW or CCW (anti-CW) direction. This was theoretically (J. Am. Chem. Soc. 2011, 133, 10, 3354–3364; Proc. Natl. Acad. Sci. USA 112 (50) 15297–15302; Proc. Natl. Acad. Sci. USA 2010 Dec 14, 107(50): 21322–21326; Proc. Natl. Acad. Sci. U.S.A. 2014, 111, 1714–1719; Proc. Natl. Acad. Sci. U.S.A. 2007, 104, 7764–7769; Proc. Natl. Acad. Sci. U.S.A. 2010, 107, 20172–20177) and experimentally (P. Nogly et al., Science 10.1126/science.aat0094 2018) shown in both animal and microbial rhodopsin rPSB chromophores.

We are aware that in chimeric channelrhodopsin C1C2 both CW and CCW isomerization pathways of the C13=C14 double bond were computed (Phys. Chem. Chem. Phys., 2015, 17, 25142–25150; Sci. Rep. 2017, 7, 7217). However, even C1C2 obeys to the above behavior since the C13=C14 displays a pre-twisting of ca. +7 degrees suggesting a CW motion and, in fact, the corresponding CW pathway was found energetically favoured with respect to the opposite CCW path.

In the revised supporting information we now provide further evidence of how the NeoR CCW isomerization is energetically disfavored consistently with the computed positive C13=C14 pre-twisting. We show that the back CCW rotation (i.e. a rotation opposite to the direction of pre-twisting) yields a steeper and less favourable S₁ profile (notice that at -180 degrees the system is only ca. 0.7 kcal mol⁻¹ lower in energy than the FC point, see Fig S4 below).

Figure S4. Comparison between the S₁ energy profiles (CASPT2/AMBER level of theory) of the first two steps associated to the CW and CCW photoisomerization pathways along the C13=C14 double bond. Orange curly arrow indicates the S₁ torsional relaxation from the Franck-Condon (FC) point.

CHANGES:

We report the discussion above and the new Fig. S4 in the revised supporting information, lines 351-363 in section S5:

“The choice of the CW rotation of C13=C14 is consistent with the *ca.* +10 degree pre-twisting seen in both the DA and the FS of the selected model. The choice is based on the observed fact¹¹⁻¹⁷ that, in both microbial and animal rhodopsins, the pre-twisting imposed by steric and electrostatic cavity effects, bias the rPSB isomerization in the CW or CCW directions. A past study on the CW and CCW C13=C14 isomerization direction in the chimeric C1C2 channelrhodopsin, showed that the CCW path displays a larger energy barrier^{17,18} consistently with a *ca.* +7 degrees CW pre-twisting. In order to exclude the possibility of a favourable CCW C13=C14 isomerization path in NeoR, in Fig. S4 we compare the initial S₁ energy profiles in the CW and CCW directions. As stated above, the pre-twisting of *ca.* +10 degrees displayed at the FC (i.e. at the DA geometry, see Fig 1A) is conserved after the small S₁ relaxation (orange curly arrow) to FS. From FS, the CCW rotation (i.e. opposite to the direction of pre-twisting) gives rise to a steeper S₁ profile (see Fig. S4)”.

Reviewer #3 (Remarks to the Author):

The authors present a computational study of the photophysical properties of the recently discovered Neorhodopsin. The whole protein environment is taken into account, and the peculiar properties of NeoR are explained in terms of environmental effects on the delocalized charge distribution of the chromophore.

The work is sound and well written and deserves to be published.

Here are my remarks

1) In the abstract, the author say that the chromophore considered features "a highly diffuse charge density". Although this will become clearer in the manuscript, this does not seem to be a very remarkable feature for a conjugated chain.

We have now made the abstract more informative by substituting the term charge density with the clearer term, charge distribution.

CHANGES:

In the abstract:

“We show that the model, that successfully replicates the relevant experimental observables, unveils a geometrical and electronic structure of the chromophore featuring a highly diffuse charge distribution along its conjugated chain.”

2) The authors claim that NeoR features a near-infrared absorption band (690 nm). As far as I know, the IR should start at 780-750 nm, so 690 is still Vis.

The reviewer is correct. We decided to avoid this issue even if, regrettably, the term “near-infrared” has been used in the original literature on NeoR (Nat Commun 11, 5682, 2020) as “a near-infrared absorbing rhodopsin”.

CHANGES:

Revised main text, lines 33-35 in the Introduction:

“In contrast, NeoR displays the strongest bathochromic shift among all known microbial rhodopsins yielding an extremely red-shifted ($\lambda^a_{max} = 690$ nm) absorption band.”

3) Very large differences in the vertical excitation energies are found by changing the protonation sites (figure 2B). Are these differences originated by a geometrical effect or by different state-specific response of the QM subsystem (or a combination of the two effects)? In other words, the large differences in the vertical excitation energies are still present if a common geometrical arrangement is used for the different protonation states considered?

In order to determine whether the differences in the absorption and emission vertical excitation energies originate from a geometrical effect or from specific interactions between protein and chromophore, we have performed additional computations. More specifically, we now compare the reported excitation energy trends of the full protein models ($\Delta E^{\text{protein}}_{S_0-S_1}$ values) with the trend computed using the isolated chromophore ($\Delta E^{\text{vacuo}}_{S_0-S_1}$ values) extracted from the protein environment and without reoptimizing its geometry. Also, we use the excitation energy of the selected a-ARM_{E141} model as a reference to quantify the extent of the trend variations. As reported in Table S4, increasing the net total negative charge of the counterion tetrad (from 0 to -3) results in a general blue shift in absorption: from the excitation energy value of 42.17 kcal mol⁻¹ displayed by the model with no counterions to excitation energy values > 70 kcal mol⁻¹ displayed by the models with 3 counterions. This is translated in energy variations ranging from ca -2 kcal mol⁻¹ (model with no counterion) to ca 38 kcal mol⁻¹ (model featuring E136 D140 E262 counterions) with respect to a-ARM_{E141}. Such behavior is strongly attenuated or even lost in vacuo where no more than ca 4 kcal mol⁻¹ (model featuring E136 E262 counterions) of variation is computed. This behavior is also found in the emission trend (Table S5).

In conclusion, the main contribution to the computed differences in vertical excitation energies among models displaying different protonation states, originates from the interaction between the chromophore (the QM subsystem) with the surrounding protein environment (the MM subsystem) displaying different protonation state configurations.

CHANGES:

We added the above discussion and the new Tables S4 and S5 in the revised supporting information, lines 144-165 in section S2:

“In order to determine whether the differences in the absorption and emission vertical excitation energies originate from a geometrical effect of the rPSB chromophore or from specific interactions between the QM and MM subsystems (or a combination of the two), we compare the reported excitation energy trends of the full protein models ($\Delta E^{\text{protein}}_{S_0-S_1}$ values) with the corresponding trends computed vacuo ($\Delta E^{\text{vacuo}}_{S_0-S_1}$ values) after extracting the rPSB chromophore geometry from its protein environment and without re-optimizing its geometry. Also, we use the excitation energy of the selected a-ARM_{E141} model as a reference to

quantify the extent of the trend variations. As reported in Table S4, increasing the net total negative charge of the counterion tetrad (from 0 to -3) results in a general blue shift in absorption: from the excitation energy value of 42.17 kcal mol⁻¹ displayed by the model with no counterions to excitation energy values > 70 kcal mol⁻¹ displayed by the models with 3 counterions. This is translated in energy variations ranging from ca -2 kcal mol⁻¹ (model with no counterion) to ca 38 kcal mol⁻¹ (model featuring E136 D140 E262 counterions) with respect to a-ARME141. Such behavior is strongly attenuated or even lost in vacuo where no more than ca 4 kcal mol⁻¹ (model featuring E136 E262 counterions) of variation is computed. This behavior is also found in the emission trend (Table S5). In conclusion, the main contribution to the computed differences in vertical excitation energies among models displaying different protonation states, originates from the interaction between the chromophore (the QM subsystem) with the surrounding protein environment (the MM subsystem) displaying different protonation state configurations.”

Table S4. Vertical energy differences (at the CASPT2/AMBER level of theory) between S₀ and S₁ at the DA computed in protein ($\Delta E^{protein}_{S_0-S_1}$) and in vacuo ($\Delta E^{vacuo}_{S_0-S_1}$). Values in parenthesis show the differences from the QM/MM model featuring E141 as rPSB counterion.

Counterion ID	$\Delta E^{protein}_{S_0-S_1}$	$\Delta E^{vacuo}_{S_0-S_1}$
No counterion	42.17 (-2.23)	41.98 (1.72)
E136	49.17 (4.77)	45.47 (1.77)
D140	50.34 (5.94)	44.80 (1.1)
E141	44.40 (0)	43.70 (0)
E262	47.46 (3.06)	45.81 (2.11)
E136 D140	78.58 (34.18)	47.22 (3.52)
E136 E141	53.05 (8.65)	45.67 (1.97)
E136 E262	65.08 (20.68)	47.68 (3.98)
E141 E262	52.42 (8.02)	45.15 (1.45)
D140 E141	54.74 (10.34)	45.78 (2.08)
D140 E262	66.39 (21.99)	45.59 (1.89)
E136 D140 E141	73.76 (29.36)	46.38 (2.68)
E136 D140 E262	82.39 (37.99)	47.49 (3.79)
E136 E141 E262	72.61 (28.21)	46.10 (2.4)
E140 E141 E262	72.14 (27.74)	46.07 (2.37)

Table S5. Vertical energy differences (at the CASPT2/AMBER level of theory) between S_0 and S_1 computed at the FS in protein ($\Delta E^{protein}_{S_0-S_1}$) and in vacuo ($\Delta E^{vacuo}_{S_0-S_1}$). Values in parenthesis show the differences from the QM/MM model featuring E141 as rPSB counterion.

Counterion ID	$\Delta E^{protein}_{S_0-S_1}$	$\Delta E^{vacuo}_{S_0-S_1}$
E136	32.05 (-9.22)	38.59 (-1.76)
D140	31.84 (-9.43)	37.69 (-2.66)
E141	41.27 (0)	40.35 (0)
E262	32.05 (-9.22)	39.00 (-1.35)
E136 D140	47.93 (6.66)	38.51 (-1.84)
E136 E141	35.08 (-6.19)	36.95 (-3.40)
E136 E262	48.00 (6.73)	38.34 (-2.01)
E141 E262	34.31 (-6.96)	36.51 (-3.84)
D140 E141	36.01 (-5.26)	35.59 (-4.76)
D140 E262	42.13 (0.86)	37.43 (-2.92)
E136 D140 E141	47.42 (6.15)	37.67 (-2.68)
E136 D140 E262	42.42 (1.15)	36.11 (-4.24)
E136 E141 E262	47.07 (5.8)	37.90 (-2.45)
E140 E141 E262	47.91 (6.68)	38.01 (-2.34)

4) In figure 2B, what is the meaning of the sentence "This model does not exhibit a DA"? The chromophore dissociates or isomerizes?

We have now realized that the presence of four counterion makes the model unstable leading to an unrealistically distorted structure. We have now removed such model.

5) In figure 3C, the two wavefunctions $[CT]+c[COV]$ and $[CT]-c[COV]$ are not orthogonal. Maybe the ground state should be $[COV]+c[CT]$?

We agree with the reviewer. We have corrected this point accordingly.

CHANGES:

We corrected the notation in Fig. 3C in the revised main text.

6) If the energies of the two diabatic states COV and CT are close, the two states experience a large diabatic coupling at the DA geometry (about 0.8 eV, i.e., half of the S_1-S_0 energy difference). This very large diabatic coupling should decrease along the isomerization coordinate, vanishing at the CoIn. The authors should comment in this respect.

CHANGES:

We now comment on this aspect in the caption to Fig. 5B in the revised main text:

“We hypothesize that the diabatic energy curves cross halfway along the isomerization coordinate α which therefore corresponds the point with the highest diabatic coupling. Such diabatic coupling will then vanish at the CoIn”

7) Figure S8: according to the caption, the map has been generated using the coordinates X1 and X2, but other coordinates are shown in the figure. Do X1 and X2 span the same space of BLA_{PSB} and alpha?

The plot is generated using the orthogonalized branching plane (BP) vectors that corresponds to the gradient difference, $\mathbf{X}_1 = \frac{\partial(V_{S1}-V_{S0})}{\partial R}$, and derivative coupling, $\mathbf{X}_2 = \langle \psi_{S0} | \frac{\partial}{\partial R} | \psi_{S1} \rangle$, where V_{S1} and V_{S0} are the adiabatic potential energy surfaces and ψ_{S0} and ψ_{S1} are the corresponding wavefunctions. Thus, our answer to the reviewer question is positive in the sense that the X1 and X2 components include both BLA_{PSB} (defined as C14-C15 and C15=N bond lengths difference) and α geometrical coordinates. Indeed, the X1 and X2 vectors represented by arrows applied to the rPSB atoms (see figure below) clearly show components of the expected BLA_{PSB} and C12-C13=C14-C15 torsional (i.e. α) motions. More specifically, it is found that X1 is dominated by α (+X1 describes a mode pointing back to the all-trans reactant) while X2 is dominated by the C15=N bond elongation/contraction (+X2 describes a strong C15-N contraction).

Figure S12. Pictorial representation of the +X₁ (left) and +X₂ (right) branching vectors at the rPSB chromophore CoIn geometry along the C13=C14 photoisomerization path.

CHANGES:

We added Fig. S12 and the above discussion in the revised supporting information, lines 482-492 in section S5.

“The plot in Fig. S13 is generated using the orthogonalized branching plane (BP) vectors that corresponds to the gradient difference, $\mathbf{X}_1 = \frac{\partial(V_{S1}-V_{S0})}{\partial R}$, and derivative coupling, $\mathbf{X}_2 =$

$\langle \psi_{S0} | \frac{\partial}{\partial R} | \psi_{S1} \rangle$, where V_{S1} and V_{S0} are the Born-Oppenheimer (adiabatic) potential energy surfaces and ψ_{S0} and ψ_{S1} are the corresponding wavefunctions. X_1 and X_2 components

include both BLA_{PSB} (defined as C14-C15 and C15=N bond lengths difference) and α geometrical coordinates. Indeed, the X_1 and X_2 vectors represented by arrows applied to the rPSB atoms (see Fig. S12) clearly show components of the BLA_{PSB} and C12-C13=C14-C15 torsional (i.e. α) motions. More specifically, it is found that X_1 is dominated by α ($+X_1$ describes a mode pointing back to the all-trans reactant) while X_2 is dominated by the C15=N bond elongation/contraction ($+X_2$ describes a strong C15-N contraction). “

Some typos:

- line 201: erase "more"
- line 256: "qualitative" should read "qualitatively"
- line 390: charge -> charged

CHANGES:

We corrected them accordingly.

REVIEWERS' COMMENTS

Reviewer #1 (Remarks to the Author):

The revision was done very properly by addressing all the points raised by the reviewers. I believe the new revised manuscript has greatly improved as well as its solidity.

I am still left with a bad feeling about the originality of the model, whose main guidelines were presented already back in the 2016 (JCTC 2016, 12, 4460). Reviewer 2 did seem to ignore this work, which justifies his/her position.

However, the authors did a great revision job. All the relevant papers have been now properly cited as well as discussed in the main text and SI.

Thus, I now recommend publication of this study on NatureComm without any further revision.

Reviewer #2 (Remarks to the Author):

In this study, Palombo et al. address the unusually high fluorescence yield of neorhodopsin, a recently discovered rhodopsin with unique spectroscopic properties. I have reviewed an earlier version of this paper where I stated that the work is indeed solid and deserves publication, but recommended to go for a more specialized journal. Since then and in view of other referee reports, the authors added numerous additional information that solidify their findings. Furthermore, they now thoroughly discuss their work in the context of previously published studies and explain the similarities and unique differences to their findings. I believe that these additions have significantly increased the impact of this work and I can now recommend publication in Nature Comm.

Reviewer #3 (Remarks to the Author):

I am satisfied with the additions and the corrections provided by the authors.

The reply to the reviewers is structured in the following way: reviewer comments (in black), author comments/reply (in blue), author proposed changes (in red).

REPLY TO THE REVIEWERS

Reviewer #1 (Remarks to the Author):

The revision was done very properly by addressing all the points raised by the reviewers. I believe the new revised manuscript has greatly improved as well as its solidity.

I am still left with a bad feeling about the originality of the model, whose main guidelines were presented already back in the 2016 (JCTC 2016, 12, 4460). Reviewer 2 did seem to ignore this work, which justifies his/her position.

However, the authors did a great revision job. All the relevant papers have been now properly cited as well as discussed in the main text and SI.

Thus, I now recommend publication of this study on NatureComm without any further revision.

We appreciate the reviewer's evaluation of the submitted revised manuscript.

As we pointed out by the reviewer, we have incorporated in the revised manuscript the description of the electrostatic constraints causing: i) a red-shifted absorption and ii) the generation of the barrier for the C13=C14 photoisomerization, in the isolated retinal chromophores (JCTC 2016, 12, 4460). In fact, in lines 423-427 and page 18 of the main text we state:

“This behavior was recently documented by El-Tahawy et al. in isolated rPSB chromophores subject to a homogeneous, strongly negative red-shifting electric fields and, therefore, the “two electron-two orbital model” theory that is shown in there to account for the electrostatic origin of the C13=C14 photoisomerization energy barrier appears also operative in the here presented NeoR¹¹.”

On the other hand, we felt that, in the interest of the readership, it was important to remain cautious and avoid making a too strong statement about the transferability of results obtained using an isolated retinal chromophore to the QM/MM model of an entire rhodopsin molecule featuring a retinal chromophore embedded in the complex molecular cavity. This decision is supported by (i) the demonstration that a simple electrostatic picture could not be sufficient to explain the fluorescence of NeoR (i.e. all S₁ barrier blocking or slowing down the isomerization), since chromophore-cavity steric effects (that, obviously, are not accounted for in the work of 2016) play an important role in the origin of the S₁ barriers and (ii) the non-homogeneity of the electrostatic field generated by the protein cavity that, most likely, has been shaped by the biological evolution to maximize the original delocalization-then-confinement mechanism reported in our manuscript.

In order to better stress these points we have inserted in the manuscript the following sentence:

CHANGES:

Revised main text, lines 48-51 in the Introduction section:

“However, it is expected that a simple electrostatic picture could not be sufficient to explain the origin of these properties in the complex environment offered by the protein cavity since other factors like non-homogeneous electrostatic fields or chromophore-cavity steric effects could play an important role.”

Reviewer #2 (Remarks to the Author):

In this study, Palombo et al. address the unusually high fluorescence yield of neorhodopsin, a recently discovered rhodopsin with unique spectroscopic properties. I have reviewed an earlier version of this paper where I stated that the work is indeed solid and deserves publication, but recommended to go for a more specialized journal. Since then and in view of other referee reports, the authors added numerous additional information that solidify their findings. Furthermore, they now thoroughly discuss their work in the context of previously published studies and explain the similarities and unique differences to their findings. I believe that these additions have significantly increased the impact of this work and I can now recommend publication in Nature Comm.

We are grateful to this reviewer for having acknowledged the impact of our findings and reconsidered the editorial destination of the work.

Reviewer #3 (Remarks to the Author):

I am satisfied with the additions and the corrections provided by the authors.

We thank the reviewer for his/her positive evaluation of our revised manuscript.